# Evaluation of the Antioxidant and Anti-Inflammatory Activities and Acute Toxicity of Caco Seed (*Chrysobalanus icaco* L.) in Murine Models

**DOI:** 10.3390/molecules29143243

**Published:** 2024-07-09

**Authors:** Abel Arce-Ortiz, Cristian Jiménez-Martínez, Gabriel Alfonso Gutiérrez-Rebolledo, Luis Jorge Corzo-Ríos, Zendy Evelyn Olivo-Vidal, Rosalva Mora-Escobedo, Yair Cruz-Narváez, Xariss M. Sánchez-Chino

**Affiliations:** 1Departamento de Salud, El Colegio de la Frontera Sur Unidad Villahermosa, Carretera Federal Villahermosa-Reforma Km 15.5, Ra. Guineo Segunda Sección, C.P., Villahermosa 86280, Tabasco, Mexico; abel.arce@posgrado.ecosur.mx (A.A.-O.); ozendy@ecosur.mx (Z.E.O.-V.); 2Escuela Nacional de Ciencias Biológicas, Instituto Politécnico Nacional, Unidad Profesional Adolfo López Mateos, Zacatenco, Av. Wilfrido Massieu Esq. Cda. Miguel Stampa S/N, Alcaldía Gustavo A. Madero, Mexico City 07738, Mexico; ggutierrezreb@gmail.com (G.A.G.-R.); rosalmorae@gmail.com (R.M.-E.); 3Unidad Profesional Interdisciplinaria de Biotecnología, Instituto Politécnico Nacional, Av. Acueducto, La Laguna Ticomán, Alcaldía Gustavo A. Madero, Mexico City 07340, Mexico; lcorzo@ipn.mx; 4Laboratorio de Posgrado e Investigación de Operaciones Unitarias, Escuela Superior de Ingeniería Química e Industrias Extractivas, Instituto Politécnico Nacional, Zacatenco, Unidad Profesional Adolfo López Mateos, Col. Lindavista, Mexico City 07738, Mexico; ycruz@ipn.mx; 5Catedra-CONAHCYT, Departamento de Salud, El Colegio de la Frontera Sur-Villahermosa, Carretera Federal Villahermosa-Reforma Km 15.5, Ra. Guineo Segunda Sección, C.P., Villahermosa 86280, Tabasco, Mexico

**Keywords:** *Chrysobalanus icaco*, bioactive compounds, antioxidants, toxicity, anti-inflammatory

## Abstract

*Chysobalanus icaco* L. (*C. icaco*) is a plant that is native to tropical America and Africa. It is also found in the southeast region of Mexico, where it is used as food and to treat certain diseases. This study aimed to carry out a phytochemical analysis of an aqueous extract of *C. icaco* seed (AECS), including its total phenol content (TPC), total flavonoid content (TFC), and condensed tannins (CT). It also aimed to examine the antioxidant and metal-ion-reducing potential of the AECS in vitro, as well as its toxicity and anti-inflammatory effect in mice. Antioxidant and metal-ion-reducing potential was examined by inhibiting DPPH, ABTS, and FRAP. The acute toxicity test involved a single administration of different doses of the AECS (0.5, 1, and 2 g/kg body weight). Finally, a single administration at doses of 150, 300, and 600 mg/kg of the AECS was used in the carrageenan-induced model of subplantar acute edema. The results showed that the AECS contained 124.14 ± 0.32 mg GAE, 1.65 ± 0.02 mg EQ, and 0.910 ± 0.01 mg of catechin equivalents/g dried extract (mg EC/g de extract) for TPC, TFC and CT, respectively. In the antioxidant potential assays, the values of the median inhibition concentration (IC_50_) of the AECS were determined with DPPH (0.050 mg/mL), ABTS (0.074 mg/mL), and FRAP (0.49 mg/mL). Acute toxicity testing of the AECS revealed no lethality, with a median lethal dose (LD_50_) value of >2 g/kg by the intragastric route. Finally, for inhibition of acute edema, the AECS decreased inflammation by 55%, similar to indomethacin (59%, *p* > 0.05). These results demonstrated that *C. icaco* seed could be considered a source of bioactive molecules for therapeutic purposes due to its antioxidant potential and anti-inflammatory activity derived from TPC, with no lethal effect from a single intragastric administration in mice.

## 1. Introduction

*Chrysobalanus icaco* L. (*C. icaco*) is a plant species distributed in both tropical America and Africa. It is part of the Chrysobalanaceae family and is known as “caco” in southeastern Mexico. This plant can adapt remarkably to diverse habitats, such as lowland jungles, mangroves, savannahs, and beach vegetation [1]. The ripe fruits of this plant are commonly used to make preserves, jellies, and refreshing drinks [2,3]. In traditional medicine in Mexico, this plant’s seeds, roots, and leaves are used to control blood glucose and body weight through infusion and decoction, among other preparations [4,5]. Moreover, this plant has been used to treat diseases, such as leukorrhea, hemorrhages, malaria, and gastrointestinal problems [6,7,8]. There is also scientific evidence of its potential use as an antioxidant agent [9].

Although there is limited information available regarding the potential benefits of the *Chrysobalanus* genus, several taxonomic species have been identified as containing secondary metabolites (SMs), specifically phenolic compounds, such as anthocyanins, gallic acid, quercetin, botulinum acid, kaempferol, and their derivatives. These compounds are found in various parts of the plant and are generally responsible for the biological activities that have been described and proven for this species, and are used in herbalism [10]. The seeds of the plant are also a rich source of essential minerals, such as potassium, magnesium, calcium, and sodium. Furthermore, terpenoids, including diterpenes and triterpenes, as well as phytosteroids, have been discovered in *Chrysobalanus* species [3,11,12].

SMs present in plants beneficially impact the various pathophysiological mechanisms by regulating the differentiation/maturation of inflammatory cells and their secretory effects [13]. This has been related to the possible prevention of non-transmissible chronic diseases (NTCD), including heart disease, atherosclerosis, rheumatoid arthritis, diabetes, cancer, and autoimmune diseases. The SMs found in *C. icaco* that have been identified with anti-inflammatory and antioxidant effects are mostly flavonoids [14], alkaloids, and phytosterols, in addition to polyunsaturated fatty acids [15], vitamins, and minerals [16].

Venancio et al. [17] evaluated the in vivo anti-inflammatory activity of a *C. icaco* fruit extract and reported a decrease in biomarkers (TNF-α, IL-1β, IL-6, and NF-kB1) at a concentration of 20.0 µg/mL GAE. Regarding previous studies on the biological activity of *C. icaco* seed and its toxicological and inflammatory effects, Ribeiro et al. [18] reported the acute toxicity of aqueous extracts of leaves with a median lethal dose (LD_50_) of >2 g/kg when administered at single doses by the intragastric route in mice. This indicates that the consumption of leaves in the form of tea could be potentially safe for humans over short periods of time (<1 week) at low quantities.

Various plant species and products are used to treat inflammatory and infectious diseases. These treatments come in the form of infusions, plasters, or creams, and are prepared using different plant parts such as leaves, stems, seeds, roots, and flowers [19]. However, most of these products lack toxicological studies to support their safety and effectiveness when used by the human population. For instance, *C. icaco* seed has not undergone any such studies to date. Therefore, the present study aimed to evaluate the potential of an aqueous extract of *C. icaco* seeds (AECS) as an antioxidant and metal-reducing agent. The study also aimed to determine the AECS’s acute toxicity (14 days) and its anti-inflammatory effect using the carrageenan-induced model of subplantar acute edema in mice.

## 2. Results

### 2.1. Phytochemical Content

Table 1 demonstrates that the AECS had a higher TPC content (124.14 ± 0.32 mg of gallic acid equivalents per g of the extract (GAE/g of the extract)) compared with the values reported for *C. icaco* seed infusions in previous studies (14.53 ± 3.80 mg GAE/g extract) [19]. However, the TPC values of Portuguese almonds (*Prunus dulcis*) and aqueous extracts of Castile walnut (*Juglans regia*) reported lower results for methanolic extracts (3.47 ± 0.17 mg GAE/g of the sample) and aqueous extracts (0.26 ± 0.55 mg GAE/g of the sample, respectively) [20,21]. These studies were compared with the present study due to the similarity in the seeds’ botanical characteristics (hard woody pericarp) and their rich essential oil content. 

The AECS had a flavonoid content of 1.65 ± 0.02 mg of quercetin equivalents per g of the extract (mg EQ/g extract) (Table 1). This value was higher than that reported for the methanolic extracts of almond trees (*Prunus dulcis*), which showed concentrations of 2.82 ± 0.11 µg catechin equivalent/g fresh weight [22]. However, in a study conducted by da Silva et al. [10], a higher TFC (6.64 ± 1.08 mg catechin equivalent/g sample) was reported for hydroalcoholic leaf extracts of *C. icaco* compared with the AECS evaluated in this study. This indicates that a greater concentration of phenolic compounds can be found in this anatomical part of this species; therefore, greater beneficial biological activity was inferred. The CT assay allows the quantification of complex SMs that have polyphenolic chemical structure such as tannins, which are commonly found in fruits as natural pigments and in the seeds of various plants, and have demonstrated a range of health benefits, including antioxidant, anti-inflammatory, antibacterial, and anticancer properties. They have also shown potential in managing obesity as an alternative treatment [23]. CTs are biosynthesized by the polymerization of flavan-3-ol units, such as catechin, epicatechin, or leucocyanidin, which are linked by a C-C bond and lack a carbohydrate nucleus [24]. The amount of CT present in an extract can vary depending on the extraction method used. In this study, CT (0.910 ± 0.01 mg of catechin equivalents per g of the extract (mg EC/g of extract)) in the AECS was found to be lower [25] than the values shown for extracts of hazelnut (*Corylus avellana* L.) using acetone–water (80:20 *v*/*v*) as extraction solvent. The authors attributed this to the strong interaction between the polarity of the solvent and that of the SM, mainly its glycosylated polyphenolic components, such as tannins. 

### 2.2. Antioxidant Potential In Vitro

After measuring the amount of SM in the AECS, the next tests focused on evaluating its antiradical properties using the DPPH and ABTS methods. Both tests showed a concentration-dependent effect, with IC_50_ values of 0.050 and 0.074 mg/mL, respectively (Table 1). Quantified compounds in the AECS, such as polyphenols, flavonoids, and tannins, have antioxidant properties, since all of them can donate electrons or hydrogen atoms to stabilize free radicals. 

A previous study evaluated the antioxidant potential of hazelnut seeds’ ethanolic extracts in vitro [26], where the researchers used a 0.1 mg/mL concentration and obtained inhibition values of 35.79 ± 0.6% for DPPH and 31.52 ± 1.3% for ABTS. However, in the current study, higher inhibition percentages were observed for the AECS at same 0.1 mg/mL concentration on DPPH (76.086 ± 0.33%), and ABTS (63.33 ± 0.58%). Another study evaluated the antioxidant potential of mature *C. icaco* seeds in vitro using DPPH at a concentration of 0.5 mg/mL and obtained an inhibition percentage of 60.1%, which was lower than the percentage obtained in this study [19]. Furthermore, in a different study, an IC_50_ of hazelnut (*Corylus avellana* L.) extracts was reported to be 1.01 mg/mL for DPPH, which is a higher concentration than that used in the current study (0.05 mg/mL) [27]. This suggests the AECS showed more significant antioxidant potential in vitro for inhibition of free radicals.

During the FRAP assay, the AECS exhibited an IC_50_ value of 0.49 mg/mL, which was lower than that reported by Locatelli et al. [28], who obtained IC_50_ values of 4.93 mg/mL and 1.01 mg/mL for acetone and methanolic extracts of hazelnut, respectively. Therefore, it can be concluded that the reducing effect of the AECS was greater. The IC_50_ values represent the minimum mg of sample/mL needed to inhibit the final concentration of the evaluated radical by 50%, or to reduce the ferric ions to a ferrous form. Thus, the lower the IC_50_, the greater is its antioxidant potential.

### 2.3. Phytochemical Profiling by FIA-ESI-FTICR-MS

Table 2 presents the primary bioactive compounds of the AECS in both positive and negative ionization modes. It is evident that nitrogenous compounds favored ionization in the positive mode, while compounds primarily containing carbon, hydrogen, and oxygen (CHO) were detected in the negative mode [29]. The detected compounds fell within a mass range (*m*/*z*) of 50 to 2000 Da (Figure 1). Ten components were identified for both modes (C_28_H_46_O_15_, C_36_H_36_O_18_, C_8_H_10_O_4_, C_38_H_48_O_22_, C_39_H_42_N_4_O_12_, C_10_H_12_N_5_O_6_P, C_10_H_12_N_4_O_6_, C_5_H_8_O, C_9_H_12_N_3_O_9_P, and C_17_H_21_N_4_O_9_P). Hence, these compounds are proposed to be present in the AECS. The compounds with a higher relative abundance were 2-acetyl-3-isobutanoyl-3,4-di(3-methylbutanoyl) sucrose, (+)-sesaminol 2-*O*-beta-d-gentiotrioside, deoxyadenosine monophosphate (dAMP), and a reduced flavodoxin, compounds that are classified as acylsugars, lignans, deoxyadenylate, and flavodoxins, respectively.

### 2.4. Acute Toxicity

During this short-term toxicological study, no abnormal changes in food and water consumption were observed in the animals. However, from 3 h onwards, groups receiving doses of 1 and 2 g/kg body weight showed unspecific signs of toxicity, such as piloerection, with frequencies of 100% (6/6 mice) and 83.33% (5/6 mice), and tachycardia (100% and 83.33%) after a single intragastric administration of the AECS. On the other hand, those receiving a dose of 500 mg/kg demonstrated behavior similar to that of the control group, starting from 4 h (Table 3).

Studies have been conducted to determine the acute toxicity of an aqueous extract of *C. icaco* leaves in both single and repeated doses over a period of 28 days. The acute toxicity results showed an LD_50_ value of >2 g/kg by the intragastric route in both male and female Swiss mice [18]. However, there is a lack of studies evaluating the toxicity of the seed extract. Ogbonnia et al. [30] evaluated the acute toxicity of *Parinari curatellifolia* seeds (belonging to the *Chrysobalanaceae* family) in Wistar rats, where the authors reported a LD_50_ value of 7.27 g/kg body weight. This showed that this seed is slightly toxic at high doses in a single administration. The LD_50_ values of both studies were greater than the 2 g/kg body weight estimated for this study. Nonetheless, according to the guidelines described by the TG423 of the OECD, doses above 2 g/kg of body weight are rejected according to the precepts of bioethics and because such doses do not emulate a single dose consumed by human beings.

Lethality is a crucial variable in evaluating the toxicity of new molecules in a preclinical stage. However, during this study, no deaths were registered within 14 days after a single intragastric administration of the AECS at the three tested doses. As a result, the LD_50_ value was greater than 2 g/kg. The increase in the body weight of the AECS-treated mice of all groups was lower than that of the healthy control group after administration of the vehicle. However, this difference was not statistically significant (*p* > 0.05), which suggests that consuming the AECS could be safe for humans in a single oral dose in lower quantities.

### 2.5. Relative Weight and Analysis of Organs (Liver, Spleen, and Kidneys)

Another way to evaluate toxicity is by analyzing the vital organs. The results of the present study showed no changes in the color and total weight of the liver, spleen, and kidneys compared with the control group. However, after calculation, the relative weight of these organs in the AECS-treated animals at doses of 2, 1, 0.5 g/kg body weight did not show any changes (*p* < 0.05) compared with the control group’s values (Table 4). This indicated that there was no increase in their size from a single intragastric administration of the AECS in healthy mice. Therefore, no macroscopic evidence was observed that indicated an inflammatory process.

### 2.6. Anti-Inflammatory Activity In Vivo

In the present study, the anti-inflammatory activity of the AECS was evaluated by a carrageenan-induced model of acute edema in the paws of mice. Table 4 presents the results where the AECS inhibited the growth of edema in the early phase of the model (1 h) by 18.14%, 32%, and 42% at doses of 150, 300, and 600 mg/kg, respectively, compared with the untreated carrageenan control group (0.62 ± 0.02 mm). These values were statistically similar to indomethacin’s in the same initial stage (32.30%). 

On the other hand, in the late stage of the model (5 h), the three doses of the AECS showed the same inhibition potential (≈53%) compared with the mice with acute edema administered only the vehicle (1.06 ± 0.04 mm). The results for the AECS-treated mice were statistically similar to those treated with indomethacin (59%) at the same time (Table 5). 

Previous studies have shown that extracts from the bark of *C. icaco* exhibited anti-inflammatory activity in murine models at three different doses (100, 200, and 400 mg/kg body weight) [30]. The results showed that at 400 mg/kg a higher percentage of inhibition of edema (53%) was generated, which was similar to that of indomethacin (69%). The results from the previous experiment are consistent with the findings of the present study.

## 3. Discussion

The seeds of diverse plants have been reported to be a rich source of dietary fiber and to contain several bioactive SMs such as phenolic acids, flavonoids, tannins, and anthocyanins, as well as sterols, tocopherols, vitamins, and minerals such as iron, selenium, and magnesium. They could act in the organism as cofactors of vital enzymes [26]. These compounds are known also for their antioxidant properties and their ability to promote regulatory physiological effects during disease-related dysregulation. They have been associated with the prevention or delay of the development of chronic degenerative diseases, such as certain types of cancer, coronary and cardiovascular diseases, obesity, diabetes, atherosclerosis, inflammation, and other diseases associated with oxidative stress [31].

Wang et al. [32] reported a significant correlation (*p* < 0.05) between seed coats and higher TPC, as well as the in vitro antioxidant potential. Therefore, the higher TPC observed in this study is likely to be due to the use of whole seeds in the extraction process. The content of SM in a sample largely depends on the solvent used and the extraction method. Advanced extraction technologies and organic compounds, with high polarity, and solvents such as methanol, ethanol, and hydroalcoholic solutions have been proven to yield higher concentrations of SM from plants after extraction [33,34,35]. Moreover, the final yield and concentration may still vary depending on the material used and the conditions of extraction [36]. This study used water as the solvent and ultrasound-assisted extraction (UAE), where ultrasonic vibrations facilitated the rupture of the cell membranes and therefore the greater release of SM. Conventional methods were not used, which can result in high shear stresses, pressure, and very high temperatures, ultimately causing the collapse of the bubbles [37]. This technique has been shown to release more easily glycosylated SMs (bound phenols), leaving the free aglycones (free phenols) to be detected and quantified more easily by in vitro techniques and increasing their possible biological activities [38].

The phenolic compounds, flavonoids, and condensed tannins present in *C. icaco* seeds depend on various factors such as the climate, the fruits’ size and stage of maturity, the variety, the methods of processing and storage, and extraction techniques [39]. Drying the seeds can prevent microbial deterioration and cause enzymatic and non-enzymatic oxidative changes that can alter the phenolic compounds’ chemical structure and their bioactivity as well [40]. Phenolic compounds contain unsaturated bonds (-C=C-) and hydroxyl groups (-OH), which give them strong antioxidant properties that help stabilize free radicals in their chemical structure [41]. However, this characteristic also makes them vulnerable to denaturation due to exposure to high temperatures, light changes in, pH, and the action of enzymes and certain metal ions, which can reduce or eliminate their antioxidant potential [42].

The TPC of medicinal plant extracts is closely linked to their antioxidant potential when tested in vitro. This is because higher TPC values indicate that the extracts can inhibit free radicals in a better manner [43]. However, it is important to note that phenolic compounds can sometimes act as pro-oxidants under specific conditions, mostly high concentrations and time of exposure [44]. On the other hand, they can also have a reducing effect on metal ions that are in their oxidized form. This can maintain or even enhance their catalytic activity, decreasing their capacity to form free radicals from peroxides, depending on the nature of the metal ions, their reducing potential, pH, and solubility [45].

The high TPC and TF in the AECS provide its well-known in vitro antioxidant potential. According to reports, polyphenolic compounds can bind with plant proteins and polysaccharides, forming covalent and non-covalent associations [46]. The presence of CT in the AECS is due to its union with fibrous cells. Some nuts (hazelnuts) are reported to contain 168 to 378 times the TPC (flavan 3-ols), while hazelnut seeds have 51.9–203.1 mg EAG/g sample, found mainly in its coat [47]. Analysis of the AECS identified compounds such as flavonoids, lignans, xanthones, aldehydes, and flavodoxin, as well as 2-acetyl-3-isobutanoyl-3,4-di(3-methylbutanoyl) sucrose, which belongs to the group of acyl sugars. The FIA-ESI-FTICR-MS technique was used to determine the molecular mass of these compounds with precision [29].

Research has shown that compounds found in the AECS, such as (+)-sesaminol 2-*O*-beta-d-gentiotrioside, xanthosine, and cyanidin 3-*O*-(6-*O*-p-coumaroyl) glucoside-5-*O*-glucoside, exhibit antioxidant activities [48], and have hypoglycemic and neuroprotective properties [49]. On the other hand, reduced flavodoxin and dAMP have anti-inflammatory effects by inhibiting NF-κB, maintaining the redox balance, and providing resistance to oxidative stress [50,51]. 

Throughout history, people have used medicinal plants extensively to treat various diseases. These important sources of pharmacological bioactive SM are used as alternative remedies in traditional medicine [52]. These plants can be used to treat both acute and chronic conditions [53]. However, conducting acute toxicity tests on murine models for plants used in traditional medicine is crucial. This is because some species may cause severe poisoning in humans that can be lethal [54]. Additionally, it has been observed that data on lethal doses in mice are more suitable for predicting toxic effects in humans [55]. To evaluate the acute toxicity according to OECD Directive 423 [56], the single-dose toxicity method was used. This method analyzed the toxicological status of the AECS depending on the mortality and morbidity status of the animals.

Body weight is also a crucial factor, as it indicates the adverse side effects of chemical substances, as it is a non-specific sign of toxicity. Studies have suggested that animals cannot lose more than 10% of their initial body weight [52]. The changes in the weight of internal organs and their relative weight ratio can help determine their normal or pathological state. In the analysis carried out on the experimental animals used in this study, there were no changes or abnormal growth in the liver, spleen, and kidneys of the AECS-treated groups compared with the control group. However, conducting a toxicological study at repeated doses for 28 days is recommended to determine whether the AECS causes changes in the relative weight of the liver, as well in the function of these vital organs. Further detailed and specific toxicological investigations of this AECS are necessary to establish its safety.

For acute inflammation caused by an injection of carrageenan, the major mechanism of action is through activation of the Bcl10, NF-kB, and IkBa pathways. This leads to the release of inflammatory mediators in two phases. In the first phase, or early phase, (1 to 3 h after injection), vasodilators such as histamine, serotonin, bradykinin, and cytokines are activated and released. Additionally, nitric oxide synthase (iNOS) is activated, and nitric oxide (NO•) is produced, increasing blood flow to the inoculation site. After this second phase (3 to 6 h), biosynthesis of prostaglandins E2 (PGE2) occurs due to the activation of induced cyclooxygenase (COX-2) enzymes, and lysosomes at 5 h, leading to the maximum formation of acute subplantar edema [57,58]. In this model, indomethacin was used, which is an anti-inflammatory drug that inhibits the enzyme cyclooxygenase (COX-2) induced in the arachidonic acid (AA) metabolic pathway. The carrageenan model involves the activation of enzymes that participate in the metabolism of arachidonic acid (AA), namely 5-lipoxygenase (5-LOX) and COX-2, with greater activity of the latter [59,60]. This model can be used to evaluate the anti-inflammatory potential of new substances on the enzymatic pathway of non-oxygen-dependent acute inflammation and biosynthesis of PGE2.

Inflammation is the body’s natural response to harmful stimuli caused by biological, chemical, and physical agents. This response can be acute, which means it occurs in the short term and is characterized by the dilation of blood vessels, increased permeability of blood vessels, swelling, migration of leukocytes and other inflammatory mediators, and self-regulation when the stimulus is eliminated, favoring the recovery of homeostasis [60]. These acute mediators include vasoactive amines, derivatives of arachidonic acid (non-oxygen-dependent pathway), and cytokines. Moreover, during this early stage of inflammation, they occur through macrophages (phagosomes) activated by reactive oxygen species (ROS) (oxygen-dependent pathway), which are intended to eliminate pathogens but can cause cellular damage in tissues and an alteration in homeostasis. Macrophages play a vital role in the initiation and propagation of acute inflammatory responses, biosynthesizing various ROS and proinflammatory cytokines (interleukin TNF-α, 1L-1β, and IL-6), leading to an accumulation of oxidative damage in tissues [61].

The excessive production of reactive oxygen and nitrogen species (ROS/RNS) leads to oxidative stress, which, in turn, causes the oxidation of biomolecules, resulting in the release of endogenous damage-associated molecular patterns (DAMPs) such as enzymes (neutral proteases, acid hydrolases, phosphatases, and lipases, among others), reactive species (superoxide, hydrogen peroxide, hydroxyl radicals, and hypochlorous acid, among others), and chemical mediators (eicosanoids, cytokines, chemokines, and nitric oxide), inducing tissue damage and oxidative stress by disrupting cellular homeostasis and triggering an inflammatory response [62,63].

To counteract oxidative damage, there are enzymatic or endogenous antioxidant systems (superoxide dismutase, catalase, and glutathione peroxidase) and non-enzymatic or exogenous antioxidant systems (Vitamin E, Vitamin C, beta-carotene, phenolic compounds, terpenes, and saponins, among others) that protect by neutralizing or eliminating reactive species, reducing the damage caused by oxidative stress, and inhibiting the inflammatory response [64].

According to this study’s findings, the AECS showed an anti-inflammatory effect that was not dose-dependent. It was attributed to the AECS’ antioxidant potential and the combined action of its different SMs, which could help remove the free radicals generated by neutrophils, macrophages, and endothelial cells during the acute inflammatory process. They may also inhibit enzymes such as COX-2 and 5-LOX that are expressed during acute inflammatory processes, thereby reducing the synthesis of PGE2 in the late phase of the experimental model, reducing the growth of edema [30]. Previous research has shown that non-steroidal anti-inflammatory drugs (NSAIDs) can inhibit both inflammatory and non-inflammatory processes when used repeatedly. NSAIDs can affect the activity of cyclooxygenase-1 (COX-1), which regulates various functions such as gastrointestinal protection, vascular homeostasis, and platelet function [13,65,66]. Therefore, future studies are required to focus not only on the isolation and evaluation of the in vitro antioxidant potential of the main SMs of the AECS and their possible anti-inflammatory effect, but also to determine the selectivity in the inhibition of COX-2 and not of the gastroprotective isoform (COX-1).

## 4. Materials and Methods

### 4.1. Biological Materials and Preparation of C. icaco Flour

Mature fruits of *C. icaco* were gathered from Paredón Bay, located in the municipality of Tonalá Chiapas, México (16°03′03″ N 93°52′00″ W). The fruits were washed and disinfected. White, cottony pulp was removed. Seeds with their coat were then dried in a convection oven at 40 °C for 72 h. They were subsequently ground into flour using a Krups grinder and sieved with a 14-mesh sieve to achieve a uniform particle size of 1.41 mm. The flour was defatted with hexane using the Soxhlet method (1:10) for 6 h, and then placed in trays at room temperature for 12 h to eliminate any excess solvent. Finally, the solvent-free dried sample was stored in polyethylene bags until use.

### 4.2. Preparation of the Aqueous Extract of C. icaco Seed

For this, 100 mg of defatted flour was weighed and added to 10 mL of distilled water, resulting in a 10 mg/mL concentration. The mixture was then placed in an ultrasonication bath (Ultrasonic Cleaner, 010S, CGOLDENWALL, Zhengzhou, China) at 42 Khz for 30 min [67,68]. After that, the mixture was filtered through a nylon filter with a pore size of 22 µm, poured into amber vials, and stored in the refrigerator at 4 °C until use.

### 4.3. Determination of Total Phenolic Content, Flavonoids, and Condensed Tannins

The following methods were used to determine the total phenol content (TPC), total flavonoid content (TFC), and condensed tannins (CT) in the AECS. 

For the TPC, a modified version of the Folin–Ciocalteu method was used, as described by Borges-Martínez et al. [69]. Gallic acid (GA) was used at concentrations ranging from 0.02 to 0.1 mg/mL (y = 5.4x − 0.0294, R^2^ = 0.9889). The AECS samples were analyzed in triplicate and read at 765 nm. The results were reported in mg of GA equivalents per gram of the extract (mg GAE/g of the extract).

A method developed by Ortega-Medrano et al. [70] was used for the TFC assay with some modifications. For this, 1 mL of the AECS was mixed with 300 µL of 5% (*w*/*v*) sodium nitrite and 300 µL of AlCl_3_ after 5 min. The mixture was allowed to stand for 6 min before 2 mL of 1 M NaOH was added. Finally, 250 µL of the mixture was deposited on microplates, and the absorbance was read at 510 nm. A standard curve of quercetin at concentrations ranging from 0.1 to 0.5 mg/mL (y = 2.4917x − 0.1017, R^2^ = 0.9961) was used to calculate the total flavonoid content. The results were expressed as mg of quercetin equivalents per gram of the extract (mg EQ/g of the extract). 

A modified version of the method described by Ortega-Medrano et al. [70] was used for measuring CT. For this, 100 μL of 1% vanillin, 100 μL of AECS, and 100 μL of 25% (*v*/*v*) sulfuric acid were mixed and incubated at 30 °C for 15 min. Themixture was read at 500 nm using a multiwell plate reader. A standard curve of catechin at concentrations ranging from 25 to 100 µg/mL (y = 0.0012x + 0.0568, R^2^ = 0.9934) was used to calculate the CT content. The results were expressed as milligrams of catechin equivalents per gram of the extract (mg EC/g of the extract).

### 4.4. Antioxidant Potential In Vitro

#### 4.4.1. DPPH Inhibition Assay

The study evaluated the antioxidant potential of the plant in vitro by inhibiting the 2,2-diphenyl-1-picrylhydrazyl (DPPH) radical, following the method described by Medina-Medrano et al. [71]. A multiwell plate reader (Multiskan Go Thermo Scientific, Waltham, MA, USA) was used for this purpose. To perform the experiment, 100 μL of the DPPH (62 µg/mL) reagent (Sigma-Aldrich D9136-5G, St. Louis, MO, USA) dissolved in ethanol was mixed with 100 μL of the AECS at different concentrations (0.02 to 0.1 mg/mL) and kept in the dark at room temperature for 20 min. The samples were then read at 517 nm, and all measurements were taken in triplicate (n = 3). The percentage of inhibition was calculated using the following formula,
DPPH inhibition (%) = [(A)blank − (A)sample/(A)blank)] × 100
where (A)blank represents the absorbance of the radical solution (DPPH solution + absolute methanol), and (A)sample is the absorbance of the test sample (DPPH solution + AECS). 

The antioxidant potential was measured in terms of IC_50_, expressed in mg/mL. The mean inhibitory concentration (IC_50_) was calculated from a graph showing the percentage of inhibition against the extract concentration and was also expressed in mg/mL. The IC_50_ represents the concentration of the sample needed to reduce the initial concentration of the radical by 50%.

#### 4.4.2. Scavenging of ABTS+ Radicals

According to Medina-Medrano et al. [71], the 2,2′-azino-bis [3-ethylbenzothiazoline-6-sulfonic acid] (ABTS) radical cation discoloration assay was performed with some modifications. To produce ABTS+, the radicals (7 mM in water) were mixed with potassium persulfate (140 mM) in the dark at room temperature for over 12 h. Before the assay, the solution was diluted in distilled water and incubated at room temperature, resulting in an absorbance of 0.70 ± 0.01 at 734 nm. The samples (0.02–0.1 mg/mL) were incubated for 20 min with 100 µL of the ABTS^+^ radical cation solution, and the absorbance was measured at 734 nm. All determinations were performed in triplicate (n = 3). The antioxidant potential was expressed in terms of IC_50_ in mg/mL. 

The percentage of inhibition of the ABTS^+^ radical was calculated using the equation below
ABTS inhibition (%) = [(A)blank − (A)sample/(A)blank)] × 100
where (A)blank represents the absorbance of the blank (ABTS+ solution + water), and (A)sample represents absorbance of the test sample (ABTS+ solution + AECS solution).

#### 4.4.3. Reducing Power

The ferric-reducing antioxidant power (FRAP) was measured according to Ramos et al. [72], using a multiwell plate reader (Multiskan Go, Thermo Scientific, USA). The AECS (75 to 300 μL at 1 mg/mL) was mixed with 250 μL of a phosphate buffer (0.2 M, pH 6.6) and 250 μL of potassium ferrocyanide (1% *w*/*v*). The mixture was incubated (50 °C/20 min), and 250 μL of trichloroacetic acid (10% *w*/*v*) was added. The mixture was centrifuged (482× *g*, 25 °C, 10 min), and 300 μL of the supernatant was taken and mixed with 300 μL of deionized water and 60 μL of ferric chloride (0.1% *w*/*v*). Then the absorbance was recorded at 700 nm. A blank reagent was prepared with distilled water instead of the AECS. The increase in the absorbance of the reaction mixture indicated an increase in the reducing power. All determinations were performed in triplicate (n = 3). The antioxidant potential was expressed in terms of the IC_50_ in mg/mL. The ferric-reducing antioxidant power was calculated using the following equation:Reducing effect (%) = [(A)sample − (A)blank)/(A)sample] × 100

### 4.5. Determination of the Phytochemical Composition by FIA-ESI-FTICR-MS Analysis

Ultrahigh-resolution mass spectra were obtained using a Solarix XR 7T (Bruker, Bremen, Germany) through flow injection analysis–Fourier transform ion cyclotron resonance mass spectrometry (FIA-ESI-FTICR-MS). The candidate metabolites’ names and structures were identified using Bruker Compass MetaboScape 2022b v. 9.0.1. The structural formulas of the compounds were retrieved from ChemSpider and the PubChem database in accordance with the conditions described by Granados-Balbuena [29].

### 4.6. In Vivo Evaluation

Female CD1 mice weighing 25 ± 5 g from Bioterio PROPECUA S.A. were used for evaluating the toxicity and anti-inflammatory effects of the studied plant in vivo. Animals were acclimatized for a week before the experiments and were kept in a controlled environment with 12-h light/dark cycles at 25 ± 2 °C and 55–80% RH. Purified water and rodent chow were provided ad libitum. Animals were cared for and maintained following the recommendations and guidelines of the International Committee for the Care and Use of Laboratory Animals (CICUAL) and was registered at https://preclinicaltrials.eu/database/view-protocol/519 (accessed on 6 July 2024) with the registration number PCTE0000519. The project was also reviewed by the Bioethics Committee of the National School of Biological Sciences of the National Polytechnic Institute (ENCB-IPN), registration number ENCB/CEI/085/2023 CONBIETICA-09-CEI-002-20190327, and El Colegio de la Frontera Sur (registration: CEI/2023/3798/05) 

#### 4.6.1. Assessment of Acute Toxicity 

According to the OECD’s test guidelines [56], this test was conducted following the TG 423 procedure. Three mice (n = 3) were randomly assigned to each of four experimental groups. Mice were fasted for 12 h prior to being administered the AECS via the intragastric (i.g.) route. The control group (Group 1) was given a single i.g. administration of the vehicle (Tween 80 and water at a 1:9 ratio) at a dose of 10 mL/kg, while Groups 1–3 were given a single i.g. administration of the AECS at doses of 500, 1000, and 2000 mg/kg, respectively. The volume of resuspended extract which was administered was 10 mL of the extract per kg of body weight. After administration, the mice were monitored for the first 6 h for any signs of neurotoxicity, such as a decrease in locomotion, aggressiveness, or reaction to a stimulus (social interactions), as well as piloerection and aspects of the feces. The frequency of the signs was recorded, along with gain in body weight (in grams) on Days 3, 7, 9, and 14 after administration. Their basal weight (Day 0) was used as a reference for comparison. Any deaths that occurred during the experiment were counted and necropsies were conducted. On the final day, the surviving mice were euthanized by cervical dislocation, and the liver, spleen, and kidneys were removed for macroscopic observation to identify pathological lesions. Finally, the relative weight (%) of these major organs was calculated [73]. As indicated in the procedural guides, depending on the number of deaths, the experiment was repeated under the same conditions and groups with three mice for each one (n = 3), until there was a total of six per group. This was carried out to establish the LD_50_ within the toxicity categories on the basis of their lethality.

#### 4.6.2. Carrageenan-Induced Model of Subplantar Edema

The potential anti-inflammatory effects of the AECS during an acute phase of inflammation were evaluated using a method described by Gutiérrez-Rebolledo et al. [73]. The study involved five groups of experimental animals (n = 7): Group (1), carrageenan control; Group (2), indomethacin (10 mg/kg); Groups (3–5), AECS at doses of 150, 300, and 600 mg/kg, respectively. Groups 2–5 were administered the treatments by the i.g. route 1 h before the irritant agent, while Group 1 received the vehicle only (Tween 80:water (1:9, 10 mL restored extract/kg)). After this, all groups (1–5) were injected with 2% carrageenan (20 µL) in isosterile saline (ISS) into the subplantar pad.

Development of subplantar edema was measured at different times after administering the irritant agent (1, 3, 5, and 7 h) using a digital vernier meter (Mitutoyo model 293–831). The formation of edema was calculated on the basis of the difference in the diameters at each hour of measurement (Td) in relation to the basal time before the injection of the irritant agent (T0). The percentage of inhibition (%) was calculated for each treatment group (2–5) by comparing them with the data from Group 1 at the same time. The results were calculated using the following formula:% inhibition = [(Td − T0) carrageenan − (Td − T0) treated/(Td − T0) carrageenan] × 100.

## 5. Conclusions

Seeds accumulate secondary metabolites, including phenolic compounds, that possess biological activities. The extraction of bioactive compounds from seeds allows us to obtain high concentrations of these compounds. The AECS from *C. icaco* seeds was found to contain high concentrations of phenolic compounds and exhibit antioxidant potential in vitro by inhibiting free radicals and reducing metal ions. Additionally, it is potentially safe for human consumption, as it did not show lethality in acute toxicity tests in CD1 mice by the intragastric route. However, further research is required to investigate the safety of repeated doses in a 28-day toxicological study. 

In an acute model induced by carrageenan at three doses, AECS decreased inflammation in the experimental animals. This could be attributed to its antioxidant potential and the different compounds contained in the extract, which work together to eliminate free radicals and inhibit enzymes during the acute inflammatory process. In future research, it is proposed that the compounds present in the AECS should be identified and the mechanisms by which the extracts regulate the inflammatory properties of other diseases related to oxidative stress should be studied.

## Figures and Tables

**Figure 1 molecules-29-03243-f001:**
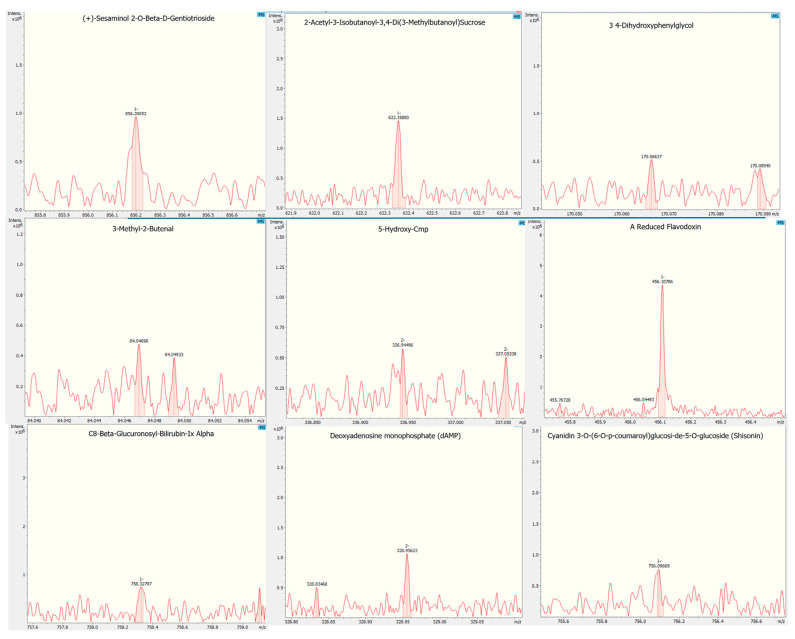
Mass spectra (MS) of the aqueous extract of *C. icaco* seed.

**Table 1 molecules-29-03243-t001:** Total content of phenolics, flavonoids, and condensed tannins, and antioxidant potential.

Assay	Content
TPC ^1^	124.14 ± 0.32
Flavonoids ^2^	1.65 ± 0.02
Condensed tannins ^3^	0.910 ± 0.01
Antioxidant techniques	**IC_50_**
DPPH ^4^	0.050
ABTS ^4^	0.074
FRAP ^4^	0.49

Data are shown as the mean (±), with its standard deviation (s.d.). ^1^ mg GAE/g of the extract; ^2^ mg EQ/g of the extract; ^3^ mg EC/g of the extract; ^4^ IC_50_, mg of extract/mL to inhibit 50% of the initial concentration of radicals, as well 50% of the reducing power. TPC, total phenolic content; IC_50_, median inhibitory concentration. Assays were carried out in triplicate (n = 3).

**Table 2 molecules-29-03243-t002:** Bioactive compounds identified in the Caco seed extract using flow injection–electrospray ionization analysis–Fourier transform cyclotron resonance spectrometry (FIA-ESI-FTICR-MS).

Theoretical *m*/*z*	Measured *m*/*z*	Formula of the Element [M − H]^+^	Error (ppm)	Compound	Relative Abundance
Positive mode					
622.3596406	622.35880	C_28_H_46_O_15_	1.350	2-acetyl-3-isobutanoyl-3,4-di (3-methylbutanoyl) sucrose	3.055
Negative mode					
756.0952107	756.09669	C_36_H_36_O_18_	1.956	Cyanidin 3-*O*-(6-*O*-p-coumaroyl)glucoside-5-*O*-glucoside (shisonin)	1.01
170.0667055	170.06637	C_8_H_10_O_4_,	1.972	3,4-Dihydroxyphenylglycol	1.080
856.1952003	856.20052	C_38_H_48_O_22_	6.213	(+)-Sesaminol 2-*O*-beta-d-gentiotrioside	2.014
758.3327712	758.32797	C_39_H_42_N_4_O_12_	6.331	C8-beta-glucuronosyl-bilirubin-IX alpha	1.56
328.9521572	328.95623	C_10_H_12_N_5_O_6_P	12.381	Deoxyadenosine monophosphate (dAMP)	2.23
284.0726800	284.07642	C_10_H_12_N_4_O_6_	13.165	Xanthosine	1.40
84.04504602	84.04698	C_5_H_8_O	23.011	3-Methyl-2-butenal	1.00
336.9370951	336.94496	C_9_H_12_N_3_O_9_P	23.342	5-hydroxy-CMP	1.210
456.1216171	456.13245	C_17_H_21_N_4_O_9_P	23.749	Reduced flavodoxin	1.516

Note: Error (ppm) was calculated as the absolute value.

**Table 3 molecules-29-03243-t003:** Frequency (%) of the appearance of behaviors of mice after a single intragastric administration of the AECS.

Time (h)	Behavioral Parameters	Experimental Groups (g/kg)			
		Control	0.5	1	2
1	Piloerection	0	17	33	33
	Lethargy	0	0	0	0
	Tachycardia	0	0	0	0
	Hyperactivity	0	17	33	17
2	Piloerection	0	50	67	83
	Lethargy	0	0	0	0
	Tachycardia	0	17	83	0
	Hyperactivity	0	0	0	83
3	Piloerection	0	67	100	83
	Lethargy	0	0	0	0
	Tachycardia	0	0	100	83
	Hyperactivity	0	0	0	83
4	Piloerection	0	17	33	83
	Lethargy	0	0	0	0
	Tachycardia	0	0	0	17
	Hyperactivity	0	0	17	0
5	Piloerection	0	17	33	50
	Lethargy	0	0	0	50
	Tachycardia	0	0	0	50
	Hyperactivity	0	0	0	0
6	Piloerection	0	0	17	50
	Lethargy	0	0	0	50
	Tachycardia	0	0	0	50
	Hyperactivity	0	0	0	0

Data are shown as the mean (±), with its standard error of the mean (SEM); the values shown are percentage frequency values considering 6/6 to be 100%. AECS, aqueous extract of *C. icaco* seed (n = 6 per group).

**Table 4 molecules-29-03243-t004:** Effect of the *Chrysobalanus icaco* L. seed extract on the relative weight of vital organs in mice.

Experimental Groups (g/kg)	Organ–Body Weight Relationship (%)		
	Liver	Spleen	Kidneys
Control	5.33 ± 0.08	0.62 ± 0.03	1.32 ± 0.03
0.5	5.28 ± 0.24	0.66 ± 0.07	1.32 ± 0.11
1	4.75 ± 0.05	0.69 ± 0.06	1.21 ± 0.05
2	4.74 ± 0.13	0.60 ± 0.05	1.33 ± 0.04

Data are shown as the mean (±) with the standard error of the mean (SEM); percentages were calculated considering the net body weight on Day 14 of the study as 100%. ANOVA statistical analysis in of ranks and Student–Newman–Keuls (SNK) post hoc test (*p* < 0.05); (n = 6 per group).

**Table 5 molecules-29-03243-t005:** Effect of the aqueous extract of *Chrysobalanus icaco* L. seeds on the diameter (mm) of acute subplantar edema in mice.

Groups (mg/kg)	Time (h)			
	1	3	5	7
Carrageenan	0.615 ± 0.02	0.751 ± 0.04 ^●^	1.056 ± 0.04 ^●▲^	0.791 ± 0.05 ^●+^
Indomethacin (10)	0.417 ± 0.04 ^a^ (32.30%)	0.285 ± 0.03 ^a●^ (62.0%)	0.463 ± 0.03 ^a▲^ (59.19%)	0.391 ± 0.04 ^a▲^ (50.38%)
AECS (150)	0.502 ± 002 ^a^ (18.14%)	0.457 ± 0.02 ^ab^ (39.14%)	0.499 ± 0.04 ^a^ (52.76%)	0.434 ± 0.05 ^a^ (45.21%)
AECS (300)	0.413 ± 0.02 ^a^ (32.89%)	0.409 ± 0.02 ^a^ (45.50%)	0.542 ± 0.07 ^a▲^ (48.68%)	0.431 ± 0.04 ^a+^ (45.69%)
AECS (600)	0.356 ± 0.02 ^ac^ (42.15%)	0.376 ± 0.03 ^a^ (49.88%)	0.445 ± 0.04 ^a^ (57.85%)	0.359 ± 0.03 ^a^ (54.64%)

Data are presented as the mean (±) and its standard error of the mean (SEM). Administration of the treatments was by the intragastric route 1 h prior to the injection of carrageenan. Values in parentheses indicate the percentage of inhibition on growth of the edema compared with the untreated carrageenan control group. Statistical analysis: ANOVA of bifactor repeated measures (RM) and the Student–Newman–Keuls (SNK) post hoc test (*p* ≤ 0.05). ^a^ Versus the carrageenan control; ^b^ vs. 10 mg/kg indomethacin; ^c^ vs. 150 mg/kg AECS; ^●^ vs. 1 h, ^▲^ vs. 3 h, ^+^ vs. 5 h. AECS, aqueous extract of *C. icaco* seeds (n = 8 per group).

## Data Availability

No new data were created or analyzed in this study. Data sharing is not applicable to this article.

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
