# Peer review of "Evaluation of the Antioxidant and Anti-Inflammatory Activities and Acute Toxicity of Caco Seed (Chrysobalanus icaco L.) in Murine Models"

_molecules, 2024, doi:10.3390/molecules29143243_

Round 1

Reviewer 1 Report

Comments and Suggestions for Authors

I have evaluated the manuscript by Abel Arce-Ortiz et al. This manuscript is interesting, but in its current form it cannot be considered further.

1. In table 2, provide the retention time value for each compound observed.

2. The compounds presented in table 2 should be tested for in silico molecular docking against receptors related to antioxidants and inflammation. It is very important to look at the pathways molecularly.

3. In the in vitro antioxidant test, what concentration gradient is used? and is there a positive control such as Trolox or Gluthatione in this antioxidant test?

4. Register this in vivo research study design at https://preclinicaltrials.eu and provide the PCT/registration number in methods.

5. Authors claim in the title "anti-inflammatory activities" but what markers or parameters demonstrate this claim? why are antioxidant and anti-inflammatory markers not checked in in vivo studies? Why?

6. Discussion of antioxidants must be connected to anti-inflammatory activities

Author Response

Dear reviewer

We appreciate the observations make on our paper; we consider that your contributions enrich our work. Below we answer your questions and comments, in the main text the changes are marked in yellow

Revisor 1

  1. In table 2, provide the retention time value for each compound observed.

We appreciate the suggestion, however the analysis was conducted by direct infusion of the sample into the mass spectrometer with resonance in the cyclotron. This technique is free of HPLC chromatography and is known as Flow Injection Analysis (FIA). FIA was chosen for its high mass accuracy and resolution. The peaks in the mass spectrometer display the m/z ratio signals of the identified molecules in their corresponding polarities (Granados-Balbuena et al., 2023; Udaeta et al., 2024).

  1. The compounds presented in table 2 should be tested for in silico molecular docking against receptors related to antioxidants and inflammation. It is very important to look at the pathways molecularly.

We appreciate your feedback. We recognize that in silico models provides a wealth of information about the potential mechanisms of action of the bioactive compounds in the extracts of Caco seeds for specific metabolic pathways, however, this is beyond the scope of our objective. Our goal was to determine whether the extract, which demonstrated antioxidant activity in vitro, could also function as an anti-inflammatory in an in vivo model.

  1. In the in vitro antioxidant test, what concentration gradient is used? and is there a positive control such as Trolox or Gluthatione in this antioxidant test?

We appreciate your observation. To conduct the antioxidant analyses in vitro, we used the IC50 parameter, which was calculated from the equation of the straight line obtained by plotting the percentage of inhibition against the different concentrations of extracts. The IC50 values were determined for DPPH (line 446-450), ABTS (line 460-461), and FRAP (line 481-482).

  1. Register this in vivo research study design at https://preclinicaltrials.eu and provide the PCT/registration number in methods.

We appreciate your observation. The protocol registration was done at https://preclinicaltrials.eu, with registration number PCTE0000519 (Line 509-510)

  1. Authors claim in the title "anti-inflammatory activities" but what markers or parameters demonstrate this claim? why are antioxidant and anti-inflammatory markers not checked in in vivo studies? Why?

We appreciate your feedback. We chose to use the carrageenan-induced supplant edema model because it is a well-studied and reproducible acute model of inflammation. This model assesses signs of inflammation such as edema, erythema, and hyperalgesia. These symptoms develop immediately after subcutaneous injection in the hind leg, leading to the production of proinflammatory agents such as bradykinin, histamine, reactive oxygen, and nitrogen species at the site of damage or infiltration into the cells. The inflammatory response can be measured by the increase in paw edema, reaching its maximum size at hour 5 after the application of the inflammatory agent (Morris, 2003).

  1. Discussion of antioxidants must be connected to anti-inflammatory activities

We appreciate your observation. In lines 355-368 the discussion between antioxidants and anti-inflammatory activity was expanded. Now it reads:

The excessive production of reactive oxygen and nitrogen species (ROS/RNS) leads to oxidative stress, which in turn causes the oxidation of biomolecules, resulting in the release of endogenous damage-associated molecular patterns (DAMPs) such as enzymes (neutral proteases, acid hydrolases, phosphatases, lipases, among others), reactive species (superoxide, hydrogen peroxide, hydroxyl radical, hypochlorous acid, among others), and chemical mediators (eicosanoids, cytokines, chemokines, nitric oxide), inducing tissue damage and oxidative stress by disrupting cell homeostasis and triggering an inflammatory response (Tan & Norhaizan, 2019 [Lines 746-747]; Truong & Jeong, 2022 [Lines 748-749]).

To counteract oxidative damage, there are enzymatic or endogenous antioxidant systems (superoxide dismutase, catalase, glutathione peroxidase) and non-enzymatic or exogenous antioxidant systems (Vitamin E, Vitamin C, beta-carotene, phenolic compounds, terpenes, saponins, among others) that protect by neutralizing or eliminating reactive species, reducing the damage caused by oxidative stress, and inhibiting the inflammatory response (Xia et al. 2022 [Lines 750-751]).

Reviewer 2 Report

Comments and Suggestions for Authors

The subject of this study was a phytochemical analysis of the aqueous extract of C. icaco seed (AECS), including its total phenols content (TPC), total flavonoids content (TFC), and condensed tannins (CT). Also, authors were examined the antioxidant and metal ion-reducing potential of AECS in vitro, as well as its toxicity, and anti-inflammatory effect in vivo in mice. Antioxidant and metal ion reducing potential was examined by inhibiting DPPH, ABTS, and FRAP. Acute toxicity test involved a single administration of different doses of AECS. These results demonstrate that C. icaco seed could be considered a source of bioactive molecules for therapeutic purposes due to its antioxidant potential and anti-inflammatory activity derived from TPC, with no lethal effect at a single intragastric administration in mice.

The work is interesting and contains a lot of results, but it is necessary to do some more experiments in order to get a complete insight into the activity of this plant.

-          In this study, only the aqueous extract of Caco seed was analyzed. Prepare another extract using polar and non-polar solvents and compare the activities of several different extracts, and then make a selection based on the obtained activities.

-          Bioactive compounds were identified in Caco seed extract using FIA-ESI-FTICR-MS. Since, in addition to identifying biologically active compounds, it is necessary to determine their content and perform HPLC analysis of the obtained extracts.

-          Abbreviations for ECG/g of extract, EQ/g of extract, and mg EC/g of extract are not introduced anywhere.

After these changes, I support the publication of this paper in the Molecules.

Author Response

Dear reviewer

We appreciate the observations make on our paper; we consider that your contributions enrich our work. Below we answer your questions and comments, in the main text the changes are marked in yellow

Revisor 2

The subject of this study was a phytochemical analysis of the aqueous extract of C. icaco seed (AECS), including its total phenols content (TPC), total flavonoids content (TFC), and condensed tannins (CT). Also, authors were examined the antioxidant and metal ion-reducing potential of AECS in vitro, as well as its toxicity, and anti-inflammatory effect in vivo in mice. Antioxidant and metal ion reducing potential was examined by inhibiting DPPH, ABTS, and FRAP. Acute toxicity test involved a single administration of different doses of AECS. These results demonstrate that C. icaco seed could be considered a source of bioactive molecules for therapeutic purposes due to its antioxidant potential and anti-inflammatory activity derived from TPC, with no lethal effect at a single intragastric administration in mice.

The work is interesting and contains a lot of results, but it is necessary to do some more experiments to get a complete insight into the activity of this plant.

  1. In this study, only the aqueous extract of Caco seed was analyzed. Prepare another extract using polar and non-polar solvents and compare the activities of several different extracts, and then make a selection based on the obtained activities.

We appreciate your recommendation. Water was used as a solvent for the extraction of bioactive compounds since this is how it is consumed by the population (e.g., infusion, decoction, crushed). In this experiment, we sought to match the traditional consumption method. However, we recognize that it would be interesting in future research to perform extractions with solvents of different polarities.

  1. Bioactive compounds were identified in Caco seed extract using FIA-ESI-FTICR-MS. Since, in addition to identifying biologically active compounds, it is necessary to determine their content and perform HPLC analysis of the obtained extracts.

Bioactive compounds were identified in Caco seed extract using FIA-ESI-FTICR-MS for precise analytical quantification by direct mass infusion, the goal was to detect and identify additional metabolites. The relative abundances of each compound were measured since the analytical technique provides data on each identified metabolite. This allowed us to determine the proportion in which these abundances are found among the metabolites and show which metabolite was most abundant compared to the rest of those identified. Now we have added a column in Table 2 with the calculated proportions (Granados-Balbuena et al., 2023).

  1. Abbreviations for GAE/g of extract, EQ/g of extract, and mg EC/g of extract are not introduced anywhere.

We appreciate the observation. Lines 93-94, 108-109 and 123 describe the abbreviations used in this document.

Round 2

Reviewer 1 Report

Comments and Suggestions for Authors

The authors have reasonably addressed the issues raised in my previous review and however, molecular docking data must be presented to strengthen the quality of this manuscript if it is considered for further publication. This is very necessary for the characterization of crude extracts and their biological activity in the 21st century.

Author Response

The authors have reasonably addressed the issues raised in my previous review and however, molecular docking data must be presented to strengthen the quality of this manuscript if it is considered for further publication. This is very necessary for the characterization of crude extracts and their biological activity in the 21st century.

We appreciate the suggestion and feedback from the reviewer. The overreaching goal of this manuscript was to characterize and quantify the compounds present in Caco seeds, as well as evaluate their antioxidant and anti-inflammatory activities. We recognize the importance of the molecular docking to predict the interaction of the compounds with different therapeutic targets. With the results from this work new research questions have arisen, including how is the interaction of the compounds contained in the extract (ligand) and a protein (receptor) and at what level? Thus, we could perform a series of in silico modeling to propose new interactions and possible therapeutic effects that we had reported in this manuscript, to obtain more robust results. Therefore, we consider that the suggestion from the reviewer is out of the scope of the present manuscript.

Reviewer 2 Report

Comments and Suggestions for Authors

The authors did not do the additional extractions that were requested but if they left it for other studies that is fine.

The assessment of the content of the detected components of the extract is included in Table 2, but it is not mentioned anywhere in the text.

Therefore, the analysis of the content of individual components in the extracts was not done.

It is really necessary to at least comment on the newly inserted results in the discussion.

Author Response

The authors did not do the additional extractions that were requested but if they left it for other studies that is fine.

The assessment of the content of the detected components of the extract is included in Table 2, but it is not mentioned anywhere in the text.

We appreciate the reviewer’s observation. In Lines 158-162 it is mentioned the compounds with the highest relative abundance.

The compounds with higher relative abundance are: 2-Acetyl-3-Isobutanoyl-3,4-Di (3-Methylbutanoyl) Sucrose, (+)-Sesaminol 2-O-Beta-D-Gentiotrioside, Deoxyadenosine monophosphate (dAMP), and A Reduced Flavodoxin, compounds that are classified as acylsugars, lignans, deoxyadenylate and flavodoxins respectively.

Therefore, the analysis of the content of individual components in the extracts was not done.

It is really necessary to at least comment on the newly inserted results in the discussion.

We appreciate the reviewer’s observation. In Lines 304-306 the compounds with higher relative abundance are mentioned, as well as in Table 2 and their reported biological activity.

On the other hand, A Reduced Flavodoxin and dAMP have anti-inflammatory effects by inhibiting NF-κB, maintaining redox balance, and providing resistance to oxidative stress.